

# Monte Carlo matrix-product-state approach to the false vacuum decay in the monitored quantum Ising chain

Jeffrey Allan Maki⋆, Anna Berti, Iacopo Carusotto and Alberto Biella

Pitaevskii BEC Center, CNR-INO and Dipartimento di Fisica,
Università di Trento, I-38123 Trento, Italy

⋆ jeffrey.maki@ino.cnr.it

## Abstract

In this work we characterize the false vacuum decay in the ferromagnetic quantum Ising chain with a weak longitudinal field subject to continuous monitoring of the local magnetization. Initializing the system in a metastable state, the false vacuum, we study the competition between coherent dynamics, which tends to create resonant bubbles of the true vacuum, and measurements which induce heating and reduce the amount of quantum correlations. To this end we exploit a numerical approach based on the combination of matrix product states with stochastic quantum trajectories which allows for the simulation of the trajectory-resolved non-equilibrium dynamics of interacting many-body systems in the presence of continuous measurements. We show how the presence of measurements affects the false vacuum decay: At short times the departure from the local minimum is accelerated while at long times the system thermalizes to an infinite-temperature incoherent mixture. For large measurement rates the system enters a quantum Zeno regime. The false vacuum decay and the thermalization physics are characterized in terms of the magnetization, connected correlation function, and the trajectory-resolved entanglement entropy.

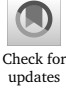

# 1 Introduction

Metastability is a ubiquitous problem in physics. This phenomenon takes place whenever a system resides at a local minimum of the (free) energy landscape (also called the *false* vacuum), which is not the the true ground state of the model (dubbed the *true* vacuum). Classically, and at zero temperature, the system will remain in the false vacuum indefinitely. Thermal fluctuations, however, could enable its decay towards the ground state configuration of the system. When quantum effects are taken into account, the system can undergo quantum tunnelling, and in an energy conserving scenario, can nucleate a resonant bubble of the true vacuum. In both cases the dynamical departure from the false vacuum is known as the false vacuum decay (FVD).

Examples of metastable systems include supercooled liquids [1], suspersaturated gases [2] and ferromagnets misaligned with respect to the magnetic field [3]. In all these examples the system is in the proximity of a first-order phase transition, but is found on the *wrong* side of the associated hysteresis loop. Such a situation can naturally be achieved by quenching a system initially in thermodynamic equilibrium across a first-order phase transition. In this non-equilibrium state, the system needs to overcome or tunnel through a potential barrier in the free-energy in order to reach a more stable state (frozen water, condensed gas, a ferromagnet correctly aligned with the magnetic field). This transition generally occurs on very long time-scales, since the two vacua are associated with two macroscopically different configurations of the system. The system is said to be in a metastable state up until the equilibrium state is reached.

For classical systems, the theory of metastability is well understood via statistical physics, where the FVD is entirely driven by thermal fluctuations. In quantum systems, both thermal and quantum flucutations can drive the FVD. The discussions of metastability driven primarily by quantum fluctuations were pioneered in the context of high-energy physics [4,5] and cosmological inflation theory [6]. Such theories describe a scenario where our universe cooled down into a metastable minimum, and could then nucleate *bubbles* of the stable vacuum via quantum tunnelling. In this scheme nucleation of bubbles of the true vacuum occurs on an exponentially long time scale. This FVD mechanism is quite general and has appeared in numerous other areas of physics [7–11]. More recently, metastable dynamics have also been

found in open quantum systems [12, 13] associated with the emergence of first-order dissipative phase transitions, and are connected to the critical slowing down [14] in bosonic [15, 16] and spin systems [17–19].

Remarkably it has recently been shown that the FVD can also be observed in one dimensional quantum spin chains [20–22]. The simplest example of such a system is the quantum Ising chain with transverse and longitudinal fields. In the ferromagnetic phase, the longitudinal field lifts the degeneracy between the two ground states with opposite magnetization. By properly tuning the system parameters, one can achieve the needed separation of the different timescales of the problem to observe the FVD.

In this class of systems, the magnetization can be used to quantify the departure from the false vacuum, and is expected to decay exponentially with time. The decay rate itself was predicted to be exponentially small in the inverse of the parameter lifting the degeneracy between the two minima [20, 22], i.e. the longitudinal field. This implies that one then has to calculate the system dynamics up to very long times in order to observe such a phenomenon in practice. Although there is agreement about the exponential behaviour of the FVD rate, there exists inconsistencies in the literature concerning the prefactor that call for further investigations [20, 22]. Furthermore, the introduction of a longitudinal field breaks the integrability of the model; no exact solution exists. For these reasons the observation of the FVD in this class of systems is extremely challenging, and its characterization remains largely unexplored, both numerically and experimentally. Only recently have works appeared in the literature characterizing the FVD [21, 23] and non-integrable dynamics [24–27] of quantum spin chains using tensor-network techniques.

Given the intrinsic quantum nature of the false vacuum decay in quantum spin chains, an important question is how its features are affected by the presence of an external measurement apparatus monitoring, for example, the local magnetization of the system. Indeed, the presence of measurements will lead to a competition between the unitary dynamics, nucleating bubbles of the true vacuum and spreading coherence, and local measurements, destroying correlations and heating up the system. Already, this interplay between coherent and dissipative dynamics at a continuous quantum transition has been shown to lead to novel physics like peculiar scaling laws in the critical regime [28, 29].

In this work we investigate the role of continuous monitoring on the physics of metastibility and FVD. We investigate this issue using a numerical approach based on the combination of a matrix product state (MPS) [30–33] ansatz for the many-body wave function and stochastic quantum trajectories [34–40]. The combination of these two techniques [38,41], which we call Monte Carlo Matrix product states (MCMPS) has recently gained an increasing amount of attention due to the possibility to study measurement-induced phase transitions in the presence of interactions [42] and the computational complexity of monitored systems [43, 44]. Crucially, this method gives access to the dynamics of single quantum trajectories. This resolution allow us to go beyond the computation of standard quantum mechanical expectation values (that could be obtained directly working with the statistical mixture generated by the stochastic dynamics) and gives the possibility to compute nonlinear quantities (as the entanglement entropy) that depends on the nature of the trajectory dynamics (and thus of the measurement protocol).

We quantify this physics from the point of view of the magnetization, two-point correlation function, and the bipartite entanglement entropy. We find that continuous monitoring of the local magnetization provides a new pathway for the system to escape the false vacuum. Our numerical results suggest that this rate is exponentially small in the inverse of the measurement rate. At the same time the monitoring also induces heating, driving the system towards infinite temperatures at long times. We analyse the typical thermalization timescale, and found signatures of the quantum Zeno effect for large measurement rates.

The paper is organized as follows: In Sec. 2 we briefly review the FVD decay mechanism in the closed quantum Ising model and present the measurement scheme. In Sec. 3 we discuss the simulation protocol used to compute the quantum trajectory dynamics within the framework of matrix-product-states. The results are then presented in Sec. 4, followed by our conclusions in Sec. 5.

## 2 The model and the measurement scheme

### 2.1 The quantum Ising chain and its false vacuum decay: A short review

The system of interest is the quantum Ising model with both transverse and longitudinal fields:

$$\hat{H} = -\sum_{i=1}^{L} \left( J\sigma_i^z \sigma_{i+1}^z + h_x \sigma_i^x + h_z \sigma_i^z \right), \tag{1}$$

where $L$ is the length of the chain, $\{\sigma_i^\alpha | \alpha = x, y, z\}$ are the Pauli matrices acting of the $i$-th site, $J > 0$ the is the nearest-neighbour ferromagnetic coupling, and $h_{x,z}$ set the magnitude of the transverse and longitudinal fields, respectively.

For $h_z = 0$, the ground state of the Hamiltonian (1) has a second-order quantum phase transition at $J/|h_x| = 1$ [45]. For $J/|h_x| > 1$, the system spontaneously breaks the inherent $\mathbb{Z}_2$ symmetry in the model ($\sigma_i^z \to -\sigma_i^z, \forall i$), resulting in a ferromagnetic phase. In this ferromagnetic phase there are two degenerate ground states with opposite local magnetization along the $z$ direction: $\langle \sigma_i^z \rangle = \pm M$ with $M = \pm(1 - h_x^2)^{1/8}$. In the regime $h_z = 0$, this system can be solved exactly by exploiting the Jordan-Wigner transformation, and thus allows for an analytical understanding of both the ground state and dynamical properties. Physically, the excitations on top of the ferromagnetic ground states are topological defects, i.e. domain walls (or kinks) interpolating between the two vacua. Since these domain walls map onto free fermionic excitations when $h_z = 0$, the energy of the system depends only on the number of kinks and their kinetic energies, not on the size of the resulting domains. Furthermore, since the fermionic excitations are non-interacting the model is integrable, hence there is no possibility for thermalization.

When $h_z \neq 0$ the situation qualitatively changes. The degeneracy between the two ground states is lifted and the energy difference between the two vacua scales extensively with the system size, $L$, as $\Delta \sim |h_z| M L$, where $M$ is the magnetization. The state where the spins are aligned with the longitudinal field (the *true* vacuum) is energetically favoured, while the state with the opposing magnetization is metastable and plays the role of the *false* vacuum. The metastability of this false vacuum depends crucially on the system's excitations. When $h_z \neq 0$, the excitations above the true vacuum can no longer be described by non-interacting fermions [46–48]. In particular, the domain walls now feel a potential linear in their separation, which prevents them from proliferating and leads to the confinement of excitations. This can be clearly seen by looking at the energy cost of forming a true vacuum bubble of size $\ell$ with respect to the false vacuum:

$$E_b = 2m - (\ell - 1)2h_z M, \tag{2}$$

where $2m$ is the energy needed to create two domain walls, while $(\ell - 1)2h_z M$ is the energy difference produced by the longitudinal field.

Since energy is conserved in the FVD process, there exists a resonant bubble size for which the energy cost vanishes: $\tilde{\ell} = 1 + m/(h_z M)$. Such a bubble can be resonantly excited during the dynamics. However, this process is very *slow* for large bubbles as the system can only virtually create bubbles of size $\mathcal{O}(1)$ until the resonant bubble of size $\tilde{l} \gg 1$ is created. Thus creating

a resonant bubble is a high-order process in $h_z$, resulting in a matrix element connecting the two states that is exponentially small in $\tilde{l} \propto 1/h_z$. In Ref. [20] the following expression for the decay rate per site has been proposed:

$$\gamma_{\text{FVD}} = \frac{\pi}{9} h_z M e^{-q/h_z}, \tag{3}$$

where $q$ and $M$ are a function of $h_x$ only. The exponential part of the decay rate (3) has been recently confirmed in numerical simulations [21].[1]

In order to observe the FVD, it is crucial $h_z/J \ll 1$ and that $h_x/J < 1$. However, $h_x/J$ can not be too close to unity, as the mass gap decreases as $h_x/J \to 1$. When this happens it is no longer justified to assume that the system wants to populate states with only two kinks (i.e. a single domain wall). When more kinks are generated there can be additional non-trivial dynamics due to the collisions of different kinks, obscuring the FVD. In Ref. [21] the authors proposed a parameter regime where the FVD could be unambiguously observed [25]. In particular for the quantum Ising chain this is found, for example, by setting $h_z/J \approx 0.08$ and $h_x/J \approx 0.4 - 0.8$.

We conclude this subsection by remarking that the numerical simulation of the FVD in quantum spin chains is computationally a hard task. Indeed, in order to probe the metastability of the false vacuum, we need to simulate the long-time dynamics following a quantum quench of an interacting spin system. Since we are dealing with a one-dimensional system, the most promising approach makes use of an infinite matrix-product-state (iMPS) ansatz for the many-body wavefunction. This ansatz accounts for the translational invariance of the system and allows to efficiently compute the time-evolved state up to times $Jt \sim 15$ for the parameters range mentioned above. This time window allows for a direct observation of the FVD, but not of the final thermalization of the system expected for $h_z \neq 0$.

## 2.2 Continuous monitoring of the quantum Ising chain: Stochastic quantum dynamics

The physics discussed previously was for the case of an isolated 1D Ising spin chain and its unitary evolution. In this work we will add a further measurement apparatus which continuously monitors the local magnetization along the longitudinal, or $z$, direction.

We note that the dynamics will depend on the particular choice of jump operators. Since the FVD physics is a phenomenon associated with the order parameter of the problem, the $+z$ component of the magnetization, we propose to measure this component of the spin. If, for example, we chose to measure the $-z$ component of the spin, the dynamics will be quite different. As the system starts in a state with a large magnetization in the $-z$ direction, the departure from the false vacuum will be suppressed, as the measurement will prefer the spins to remain oriented along the $-z$ direction. On the other hand, if we measure the spin along, say, the $+x$ direction, one would expect different dynamics.

When the $+z$ component of the spins of the quantum Ising model are measured continuously in time, the evolution of the many-body wavefunction is governed by quantum trajectories $|\psi(\mathbf{N_t})\rangle$ which follow the following stochastic Schrödinger equation [50]:

$$d|\psi(\mathbf{N_t})\rangle = dt \left[ -iH - \frac{\gamma_d}{2} \sum_{i=1}^{L} \left( L_i^\dagger L_i - \langle L_i^\dagger L_i \rangle_{\mathbf{N_t}} \right) \right] |\psi(\mathbf{N_t})\rangle + \sum_{i=1}^{L} \left( \frac{L_i}{\sqrt{\langle L_i^\dagger L_i \rangle_{\mathbf{N_t}}}} - 1 \right) \delta N_t^i |\psi(\mathbf{N_t})\rangle, \tag{4}$$

---

[1]The prefactor in Eq. (3) is non-universal, and is currently debated in the literature. See e.g. Ref. [22]. Furthermore, by tuning the ratio between the longitudinal $h_z$ and the transverse field $h_x$, one could activate new more relevant decay paths giving rise to different decay behaviours [49].

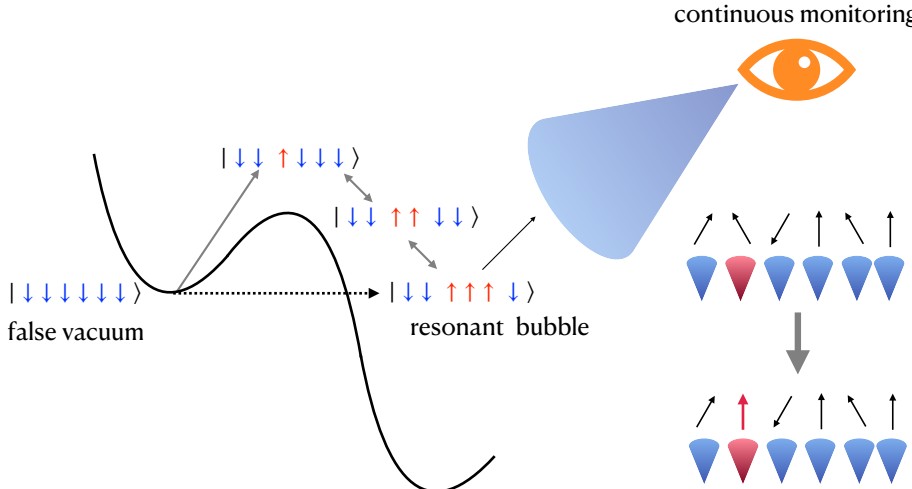

Figure 1: A sketch of the system under consideration. The decay of the false vacuum of the quantum Ising chain takes place through the virtual occupation of $\mathcal{O}(\tilde{l})$ off-resonant states ($\tilde{l}$ being the size of the bubble). This metastable dynamics is continuously monitored by measuring the local magnetization. This process induces incoherent spin flips in random positions (denoted by red spins) and affects the closed-system dynamics.

where $H$ is the system Hamiltonian (1) ruling the unitary evolution, $\gamma_d$ is the measurement rate, and the jump operators are denoted by $L_i$. For this work we consider jump operators that measure the $+z$ component of the spin:

$$L_i = n_i \equiv \frac{\sigma_z + \mathbb{I}}{2}. \tag{5}$$

In Eq. (4), we also define the vector $\mathbf{N_t} = [N_t^1, N_t^2, \ldots, N_t^L]$ is a collection of uncorrelated Poisson processes which satisfy $\delta N_t^i = 0, 1$, $\left(\delta N_t^i\right)^2 = \delta N_t^i$ and have expectation values $\mathbb{E}[\delta N_t^i] = \gamma_d dt \langle L_i^\dagger L_i \rangle_{\mathbf{N_t}}$ where $\langle \bullet \rangle_{\mathbf{N_t}} = \langle \psi(\mathbf{N_t}) | \bullet | \psi(\mathbf{N_t}) \rangle$. The vector $\mathbf{N_t}$ parametrizes when and where the quantum jumps occur in the course of the dynamics. Thus a given vector $\mathbf{N_t}$ describes a given experimental realization of the system.

The quantum jump trajectories, or simply quantum trajectories (QTs) of Equation (4) faithfully describe the dynamics when the monitoring apparatus acts occasionally but abruptly on the system causing a random local spin to be projected along the $+z$ direction (second term in Eq. (4)) with probability $p_i(t) = \mathbb{E}[\delta N_t^i] = \gamma_d dt \langle n_i \rangle_{\mathbf{N_t}}$ proportional to the measurement rate and to the probability of the spin on the $i$-th site to be in the $+z$ direction. When no jump occurs, the system evolves according to the non-Hermitian Hamiltonian (the first term in Eq. (4)):

$$H_{\text{eff}} = H - i\frac{\gamma_d}{2}\sum_{i=1}^{L} L_i^\dagger L_i = H - i\frac{\gamma_d}{2}\sum_{i=1}^{L} n_i, \tag{6}$$

with probability $1 - \sum_{i=1}^{L} p_i(t)$.

For simplicity, we label the wavefunction resulting from a single noise realization as $|\psi_\alpha(t)\rangle$ and the conditional density matrix $\rho_\alpha(t) = |\psi_\alpha(t)\rangle\langle\psi_\alpha(t)|$. From these quantities we can

reconstruct the mean state of the system at a given time $t$ as:

$$\overline{\rho}(t) = \lim_{N_{\text{traj}} \to \infty} \frac{1}{N_{\text{traj}}} \sum_{\alpha=1}^{N_{\text{traj}}} \rho_\alpha(t), \tag{7}$$

where $N_{\text{traj}}$ is the number of QTs. One can readily show that given the stochastic Schrödinger equation in Eq. (4), the equation of motion for the mean density matrix $\overline{\rho}(t)$ is the Linblad master equation [51]:

$$\frac{d}{dt}\overline{\rho}(t) \equiv \mathcal{L}[\overline{\rho}(t)]$$
$$= -\frac{i}{\hbar}[H_{\text{eff}}, \overline{\rho}(t)] + \gamma_d \sum_{i=1}^{L} n_i \rho(t) n_i, \tag{8}$$

where we have defined the Liouvillian superoperator $\mathcal{L}[\bullet]$. From Eq.(8) we can conclude that the average dynamics induced by the continuous monitoring of the local magnetization is equivalent to that of a system coupled to an infinite temperature thermal bath causing pure dephasing at a rate $\gamma_d$.

Equation (8) admits a unique stationary state that is the maximally mixed density matrix:

$$\rho_{\text{ss}} \equiv \lim_{t \to \infty} \overline{\rho}(t) = \frac{\mathbb{I}}{\mathcal{D}}, \tag{9}$$

where $\mathcal{D}$ is the dimension of the relevant Hilbert space. If the system dynamics is constrained by some symmetries of the model the relevant Hilbert space spanned by the dynamics may be smaller than the full Hilbert space, depending on the initial conditions. In our case, the presence of continuous measurements breaks all symmetries (for $|h_x| \neq 0$ regardless of $h_z$), so that the system flows to the diagonal ensemble of the full Hilbert space and $\mathcal{D} = 2^N$. This can be easily seen by noting that the non-Hermitian Hamiltonian of the model (6) possesses a complex-valued longitudinal field with amplitude $h_z - i\gamma_d/4$. In other words, the continuous measurement protocol heats up the system, asymptotically driving it toward an equally-probable incoherent mixture of all the allowed many-body states, which is what we denote as infinite temperature. At large times the mean state is expected to relax exponentially to $\rho_{\text{ss}}$, and the typical relaxation rate $\gamma_{\text{th}}$ would be given by the so-called Liouvillian gap (i.e. the spectral gap of $\mathcal{L}$), characterising the asymptotic decay rate of the system [14].

The quantum expectation values of generic quantities, $O$, which are independent of the state of the system $\rho_\alpha$ are related to the average over QTs:

$$\langle O \rangle(t) = \text{Tr}[O\overline{\rho}(t)] = \lim_{N_{\text{traj}} \to \infty} \frac{1}{N_{\text{traj}}} \sum_{\alpha=1}^{N_{\text{traj}}} \langle \psi_\alpha(t)|O|\psi_\alpha(t) \rangle, \tag{10}$$

i.e. we can sample the expectation value of a given observable by averaging over many stochastic realization.

However, if the quantity, $O$, we want to compute depends on $\rho_\alpha(t)$, the second equality in (10) does not hold. A particular example relevant to our case is the bipartite entanglement entropy:

$$S_\alpha(t) = -\text{Tr}[\rho_\alpha^A(t) \ln \rho_\alpha^A(t)], \tag{11}$$

where the reduced density matrix for region $A$ is $\rho_\alpha^A(t) = \text{Tr}_B[\rho_j(t)]$, with $\text{Tr}_B[\bullet]$ denoting the partial trace over the complimentary region $B$. One can immediately see that the average of Eq. (11) over quantum trajectories:

$$S(t) = \lim_{N_{\text{traj}} \to \infty} \frac{1}{N_{\text{traj}}} \sum_{\alpha=1}^{N_{\text{traj}}} S_\alpha(t) \neq -\text{Tr}[\overline{\rho}^A(t) \ln \overline{\rho}^A(t)], \tag{12}$$

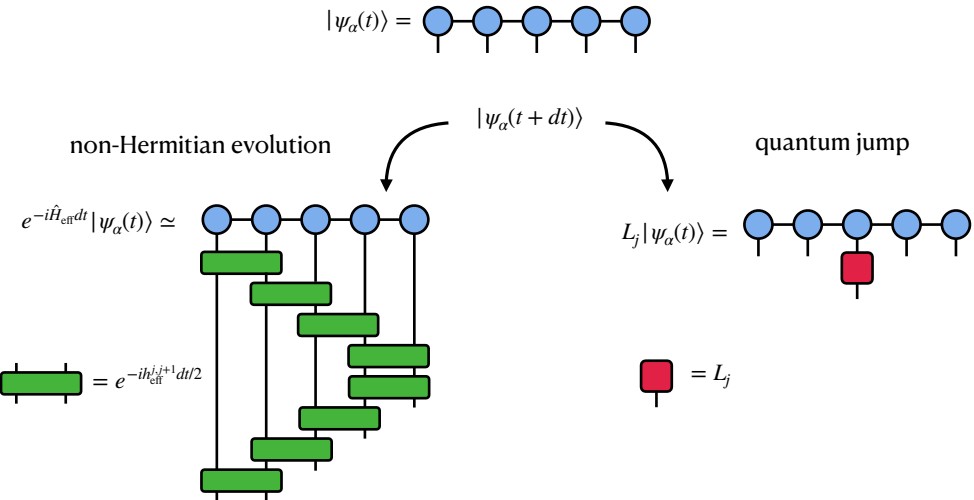

Figure 2: A sketch of the MCMPS method. At each time step $dt$ the evolution of the MPS $|\psi_\alpha(t)\rangle$ obeys the stochastic dynamics (4). With probability $1 - \sum_{i=1}^{L} p_i(t)$ and $p_i(t) = \gamma_d dt \langle n_i \rangle_\alpha$, the system evolves according to the effective non-Hermitian Hamiltonian $H_{\text{eff}}$ in its Trotterized form (16) (left side) or, with probability $\sum_{i=1}^{L} p_i(t)$, undergoes a quantum jump. In this a quantum jump occurs on the $j$-th site (right side).

is not the same as the entanglement entropy one would obtain from using the mean reduced density matrix over the subspace $A$, $\overline{\rho}^A(t)$. The entanglement entropy calculated from the reduced density matrix will contain classical contributions due to the fact that $\overline{\rho}^A(t)$ is a mixed state, alongside the contributions from quantum entanglement. For this reason the entanglement entropy $S$ is a quantity that depends on the specific trajectory protocol arising from a given measurement procedure.

In both cases, we can evaluate the statistical error at a given time $\sigma(t)$ in the trajectory mean by computing the standard deviation of that distribution. This error is found to be [38]:

$$\sigma(t) = \frac{\Delta X(t)}{\sqrt{N_{\text{traj}}}}, \tag{13}$$

where $\Delta X(t)$ is the standard deviation of the trajectory outcomes distribution at a given time $t$ for the quantity $X$. For outcomes distribution with non-zero mean $X_{\text{mean}}$ we chose $N_{\text{traj}}$ such that $\sigma(t)/X_{\text{mean}} \ll 1$ (which implies $N_{\text{traj}} \gg (\Delta X(t)/X_{\text{mean}})^2$).

Our approach based on MCMPS allows us to simulate the dynamics of individual QTs, thus enabling the study of both trajectory-dependent nonlinear quantities (like the entanglement entropy in (12)) as well as the quantum expectation value of standard observables like the magnetization (as described in (10)). Provided we perform the simulations a large number of times, we can obtain results with small statistical uncertainties. A sketch of the system under consideration is shown in Fig. (1).

## 3 Simulation protocol with Monte Carlo matrix product states

In this work we numerically compute the system dynamics according to the stochastic Schrödinger equation, Eq. (4), after that the system is initially prepared in the false vacuum.

To this end we adopt a MPS representation of the many-body state [30], and we evolve the wavefunction using the Time Evolving Block Decimation (TEBD) scheme [32] combined with stochastic QTs accounting for the measurement process [41]. This method goes under the name of Monte Carlo matrix product states (MCMPS). All numerical calculation were done using the ITensor library [52,53] of the Julia Programming Language [54].

The main steps of the algorithm are summarized as follows:

- **Ground state preparation**. We first prepare the system in the ground state of the Hamiltonian in Eq. (1) with fields $h_x$ and $h_z$, $H(h_x, h_z)$. This is done using an imaginary time evolution starting from an initially random MPS of $L = 100$ sites. The imaginary time evolution is also done using the TEBD scheme with a cutoff of singular values set to $10^{-8}$, which controls the truncation error for the state propagation. We evolved the system up until an imaginary time $J\tau = 10$ with an imaginary time step $Jd\tau = 10^{-2}$. This choice of parameters provided adequate convergence.

- **Quench from the false vacuum**. From the initial state, we suddenly quench the longitudinal field globally: $h_z \to -h_z$, and evolve the initial state according to the same stochastic Schrödinger equation, Eq. (4), but with the Hamiltonian to $H(h_x, -h_z)$. If the magnitude of the longitudinal field $h_z$ is small compared to the other energy scales this procedure can be seen as a quench to the the false vacuum of the Hamiltonian $H(h_x, -h_z)$. However this procedure always produces some unwanted low-lying excitations on top of the false vacuum that will affect the short-time behavior of the system.

- **Stochastic quantum dynamics**. The algorithm for implementing Eq. (4) was shown in Refs. [38,41], In order to implement the stochastic dynamics in Eq. (4), we discretize the time evolution and after each time step, $dt$, we stochastically choose whether to evolve the system with the non-Hermitian effective Hamiltonian (6) [with probability $1 - \sum_{i=1}^{L} p_i(t)$]:

$$|\psi_\alpha(t+dt)\rangle = \frac{e^{-iH_{\text{eff}}dt}|\psi_\alpha(t)\rangle}{\|e^{-iH_{\text{eff}}dt}|\psi_\alpha(t)\rangle\|}, \tag{14}$$

or, otherwise, to apply the $i$-th jump operator [with probability $p_i(t)$]:

$$|\psi_\alpha(t+dt)\rangle = \frac{n_i|\psi_\alpha(t)\rangle}{\|n_i|\psi_\alpha(t)\rangle\|}. \tag{15}$$

The trajectory evolution scheme described above has to performed within the MPS representation of the many-body wavefunction. The non-Hermitian evolution ruled by the effective Hamiltonian in Eq. (6) can be easily cast into a MPS friendly form using the Trotter decomposition:

$$e^{-idtH_{\text{eff}}} \simeq \left(\prod_{i=1}^{L-1} e^{-ih_{\text{eff}}^{i,i+1}dt/2}\right)\left(\prod_{i=1}^{L-1} e^{-ih_{\text{eff}}^{L-i,L-i+1}dt/2}\right) + \mathcal{O}\left(dt^3\right). \tag{16}$$

In defining Eq. (16) we used the fact that the effective Hamiltonian contains only local and nearest neighbours terms and thus can be written as

$$H_{\text{eff}} = \sum_{i=1}^{L-1} h_{\text{eff}}^{i,i+1}, \qquad h_{\text{eff}}^{i,i+1} = -(J\sigma_i^z\sigma_{i+1}^z + h_x\sigma_i^x + h_z\sigma_i^z). \tag{17}$$

The action of a given quantum jump can be easily computed by applying the local operator $L_i = n_i$ to the MPS structure. The whole MCMPS procedure is illustrated in Fig. (2).

As one may expect, the weak continuous measurement protocol is very sensitive to the time step $dt$. From our explorations[2] we found the optimal time step to be $Jdt = 10^{-3}$. Unless otherwise specified, we consider systems of size $L = 100$, and work with an initial Hamiltonian with $h_x/J = 0.8$ and $h_z/J = 0.08$. This choice of parameters was used in Ref. [21] to observe the FVD in the absence of measurement, and provides a benchmark against the closed system.

## 4 Results

In this section we present the numerical results obtained with the MCMPS method. To characterize the false vacuum decay we consider several observables that allow us to draw a clear picture of the dynamics. The emerging scenario highlights two different regimes: At short and intermediate times the departure from the local minimum is accelerated by the monitoring while, at long times, the system thermalizes to an infinite-temperature incoherent mixture. We start our analysis considering the dynamics of the monitored local magnetization.

### 4.1 Magnetization and metastability of the false vacuum in the presence of measurements

In the analysis of the false vacuum decay one of the main observables routinely utilized in the literature (see e.g. Ref. [21]) is the magnetization fidelity:

$$F(t) = \frac{\sum_{i=1}^{L} \left( \langle \sigma_i^z(t) \rangle_t + \langle \sigma_i^z(0) \rangle_t \right)}{2 \sum_{i=1}^{L} \langle \sigma_i^z(0) \rangle_t} . \tag{18}$$

The magnetization fidelity explicitly quantifies the departure from the false vacuum, i.e. when $F(0) = 1$. In [21] has been shown how this quantity decay exponentially with a rate that follows Eq. (3). In order to understand the effect of the continuous monitoring we compute, for the same value of the Hamiltonian parameters, the dynamics of $F(t)$ for various values of $\gamma_d$. The results are shown in Fig. (3).

Fig. (3) allows us to identify several important features in the dynamics. After an initial transient (lasting up to $Jt \sim 1$) we generally observe two main regimes. The first is an intermediate time regime associated with the FVD where $F(t)$ still decays exponentially with a rate that now depends also on $\gamma_d/J$. The second main regime emerges at long times where the exponential decay associated to the FVD progressively disappear and the system approaches the infinite temperature state, (9), which has vanishing magnetization along all directions and thus leading to $\lim_{t\to\infty} F(t) = \frac{1}{2}$. These two regimes are dictated by two unique and independent frequency scales: $\gamma$ describing the FVD at intermediate times, and $\gamma_{\text{th}}$ describing the long-time thermalization. In the following discussions we will characterize the intermediate time FVD dynamics and the long-time thermalization dynamics not only using the magnetization, but also the two-body correlation function and entanglement entropy.

#### 4.1.1 Intermediate times: Measurement-induced acceleration of the FVD

First, let's consider the dynamics of the fidelity at intermediate times where we observe the FVD, an exponential decay of the magnetization fidelity. For each value of $\gamma_d$, we extract the FVD rate, $\gamma$, by fitting $F(t)$ to an exponential decay within the appropriate time-window. The details of this procedure are shown in Appendix A, while the results are shown in Fig. (4).

---

[2]For any larger time steps, we observed discrepancies in the quantum trajectories for $Jt \gtrsim 10$. The quantities under consideration are always averaged over $N_{\text{traj}} \geq 600$.

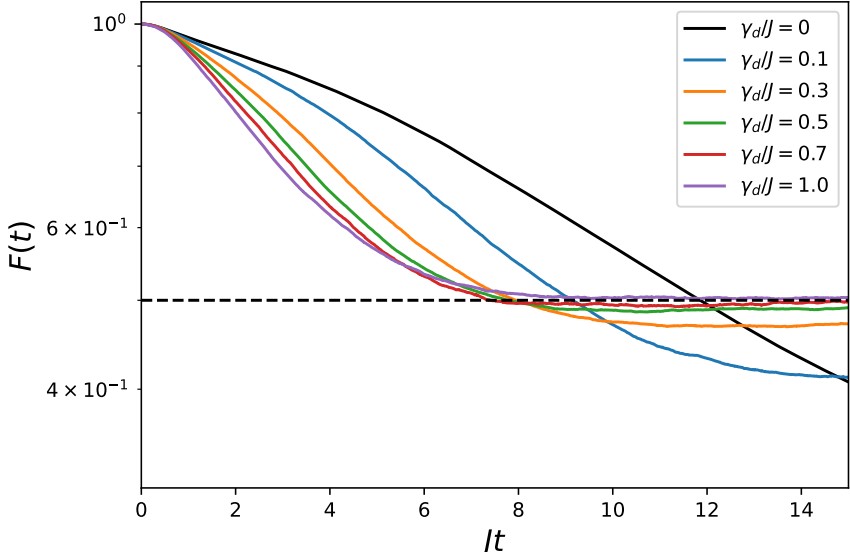

Figure 3: $F(t)$ defined in Eq. (18) for various values of the coupling to the environment, $\gamma_d$. In these simulations we consider a system of size $L = 100$, and with parameters $h_x = 0.8$ and $h_z = 0.08$. Each solid line represents the average of $N_{\text{traj}} \geq 600$ trajectories. The dashed line corresponds to the infinite temperature steady state where $F(t) = 1/2$. The statistical error (13) is such that $\sigma(t)/F(t) \lesssim O(10^{-2})$ at all times and thus it is too small to be visible on this scale.

Inspired by the analytical formula for the FVD rate in a closed system, we fit the numerical results for the FVD rate, $\gamma$, to a phenomenological Arrhenius law:

$$\gamma \propto \exp\left[-\frac{AJ}{B\gamma_d + h_z}\right], \tag{19}$$

The fit appears to describe the physics reasonably well[3] for the range of measurement rates considered. Equation (19) is appealing as it smoothly connects to the expression (3) for vanishing measurement rate $\gamma \to 0$[4] and states that the departure from the false vacuum is exponentially small in $1/\gamma_d$ up to $\gamma_d \sim J$.

The fact that the FVD decay rate is still exponentially suppressed for quite large values of $\gamma_d$ is quite surprising. It suggests that the metastability of the false vacuum is not immediately spoiled by measurements: The coupling to the environment assists the tunneling process and renormalizes the decay rate, i.e. the general trend remains the same. This is even more striking since the mechanism for departing from the false vacuum is quite different in the monitored scenario; the measurements can make a a single site with virtual spin in the $+z$ direction real at a rate $\gamma_d$. This process then causes a cascade of further measurements as the probability for a measurement to occur is proportional to $\langle n_i \rangle_\alpha$, i.e. the probability for a spin to be oriented along the $+z$ direction.

To further study this mechanism, we examined the local magnetization for a single QT for various $\gamma_d$, see Fig. (5). When $\gamma_d = 0$, we see the magnetization evolves slowly in the bulk. There are also significant dynamics in the magnetization at the boundaries due to finite size effects. When $\gamma_d \neq 0$, we see that first, the change in the magnetization in the bulk is slower than when $\gamma_d = 0$. This is due to the non-Hermitian evolution of the system which

---

[3]with $A \approx 0.07$ and $B \approx 0.3$.
[4]For $\gamma_d = 0$ our results slightly differ quantitatively with respect to what reported in Ref. [21]. This is due to finite size effects which are discussed in Appendix B.

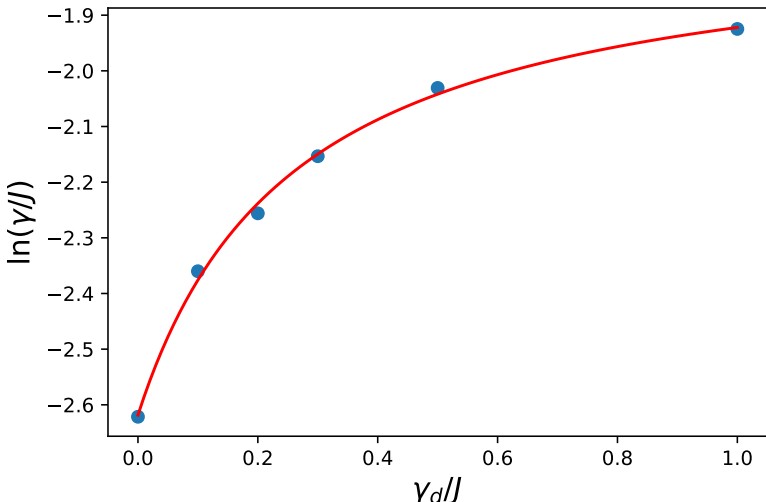

Figure 4: FVD rate, $\gamma$, as a function of $\gamma_d/J$. The solid dots correspond to the results of the QT simulation, and the red line is a fit to Eq. (19).

favors the spins to stay oriented in the $-z$ direction and suppresses the states with spins in the $+z$ direction. A nice consequence of the non-Hermitian evolution is that finite size effects do not penetrate into the bulk, and one can access the thermodynamic limit more quickly. This is discussed in more detail in Appendix B. The initial change in the magnetization primarily comes from the measurement process creating a local spin oriented along the $+z$ direction. These excitations then expand ballistically, causing more measurements. We do not observe the confinement of excitations on this time scale for finite $\gamma_d$ due the cascade of further measurements. For larger values of $\gamma_d$ this process occurs at a larger rate thus driving faster the system away from the false vacuum.

Since measurements are the leading mechanism driving the system away from its initial state, it is quite natural to expect the same physics to occur in the limit of zero longitudinal field $h_z = 0$, i.e. the transverse Ising model. In this case, we study the dynamics when the system is prepared in the ground state where the magnetization is in the $-z$ direction. The measurement apparatus can still project local spins onto the $+z$ direction, which starts a cascade of further measurements that melts the order in a manner similar to the case of finite $h_z$. Thus we expect there is an exponential decay in $F(t)$ with a decay rate, $\gamma$, given by Eq. (19) but with $h_z = 0$. We have numerically confirmed that the melting of the order exhibits an exponential decay that is described by an Eq. (19), as discussed in Appendix C.

### 4.1.2 Long times: Thermalization and the emergence of the quantum Zeno regime

The second major feature of the dynamics of the magnetization is the decay towards the infinite temperature state at long times. The asymptotic decay ($tJ \gg 1$) towards $\rho_{ss}$ is exponential with a thermalization rate, $\gamma_{th}$:

$$\|\overline{\rho}(t) - \rho_{ss}\| \sim e^{-\gamma_{th}t}, \tag{20}$$

which implies $|F(t) - 1/2| \sim e^{-\gamma_{th}t}$ for $Jt \gg 1$. For the parameters under consideration, we witness thermalization for $\gamma_d/J \sim 1$. For significantly smaller or larger values of $\gamma_d/J$ the thermalization time scale is longer than the time scales accessible to our MPS calculation.

To overcome these numerical limitations, we note that the thermalization rate must correspond to the spectral gap of the Liouvillian superoperator defined in Eq.(D.1); the thermal-

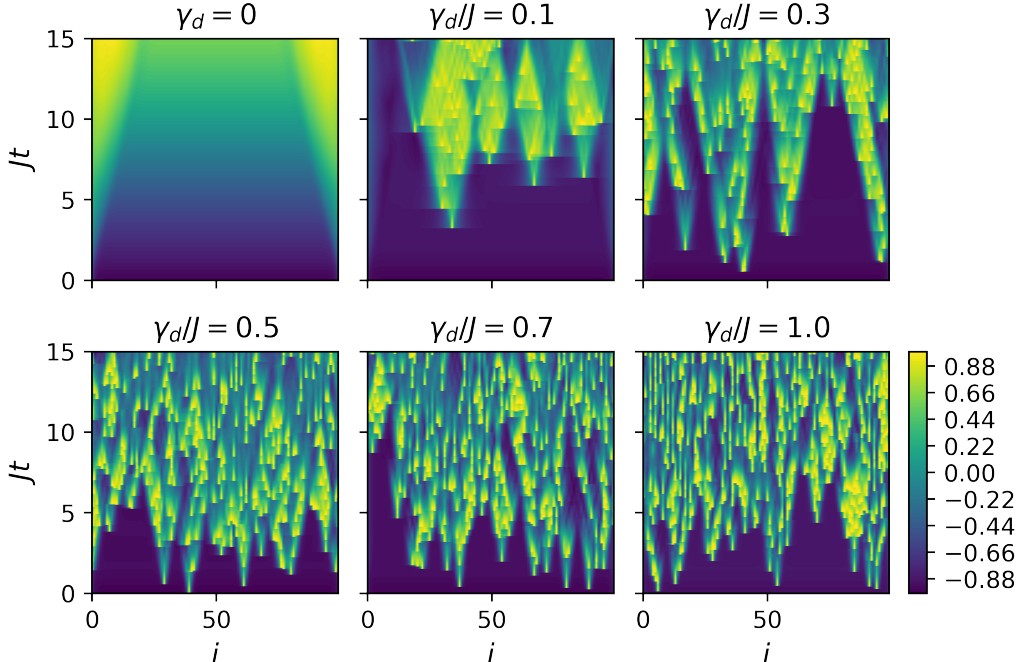

Figure 5: Local magnetization for a single quantum trajectory for various $\gamma_d$. For $\gamma_d = 0$, the change in magnetization is dominated by the spins at the boundary, and represent finite size effects. For finite and increasing $\gamma_d$ one observes the appearance of a single spin projected along the $+z$ direction due to a quantum jump. This single site domain wall appears to spread ballistically and causes further quantum jumps, nucleating more spins. The number of quantum jumps increases both as a function of time and of $\gamma_d$.

ization rate is governed by the eigenvalue of $\mathcal{L}$ with the smallest absolute value of the real part [14]:

$$\gamma_{\text{th}} = -\text{Re}\left[\lambda_1\right]. \tag{21}$$

Hence the thermalization rate can be accessed by diagonalizing the Liouvillian superoperator.

In Fig. (6) we report the thermalization rate for a quantum Ising spin chain obtained via exact diagonalization for a system of size $L = 6$ for and various values of $\gamma_d/J$. The values of the longitudinal and transverse fields, $h_z$ and $h_x$, are the same as those used in the simulations shown in Fig. (3). As one can see for small $\gamma_d/J$, the value of $\gamma_{th}$ increases with measurement rate $\gamma_d$. This intuitive behaviour indicates that the faster the system is monitored, the faster the chain heats up toward $\rho_{\text{ss}}$. However, for $\gamma_d/J \gtrsim 5$ we find that the thermalization rate decreases with increasing $\gamma_d$. This signals the appearance of the quantum Zeno regime [55–58] in our protocol. In this regime the system is governed by a reduced subspace of dark states which are insensitive to the monitoring. In our case such dark states correspond to density matrices with definite magnetization along $z$, see Appendix D.

In the limit $\gamma_d \gg h_x$ (defining the quantum Zeno regime of the model and that in our case also implies $\gamma_d \gg J$) we can obtain an analytical expression for $\gamma_{\text{th}}$ by employing a dissipative Schrieffer-Wolff transformation [59] in order to construct an effective Liouvillian for these dark states. The details of this calculation are shown in Appendix D. The result is that the thermalization rate in the quantum Zeno regime is given by

$$\gamma_{\text{th}} \approx \frac{8h_x^2}{\gamma_d}. \tag{22}$$

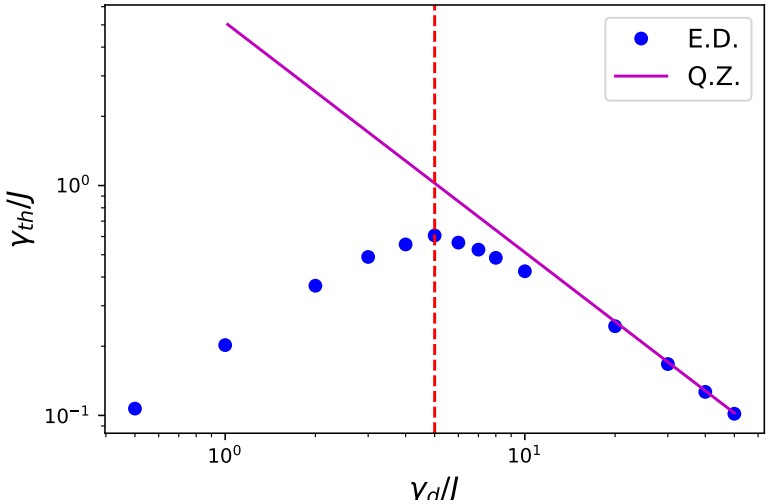

Figure 6: The Liouvillian gap $\gamma_{\text{th}}$ as determined from exact diagonalization (E.D.) of the Liouvillian for a small system of $L = 6$ alongside the analytical prediction in the Quantum Zeno (Q.Z.) regime, Eq. (22). We consider $h_x/J = 0.8$ and $h_z/J = 0.08$. For these parameters, the transition to the Q.Z. regime is denoted by the red dashed line, and occurs for $\gamma_d/J \approx 5$.

Equation (22) is independent of the system size, and applies equally to infinitely large systems as local processes dominates over the non-local coupling rate $J$ in the quantum Zeno regime. One key feature to note is that Eq. (22) doesn't depend on either $J$ or $h_z$ to leading order, which is a consequence of the fact that we monitor the $z$ component of the spin.

In Fig. (6) we also present this analytical solution alongside the thermalization rate obtained from the exact diagonalization of the Liouvillian. We find excellent agreement for large $\gamma_d/J$. For small values of $\gamma_d/J$ we find that the thermalization rate is proportional to $\gamma_d/J$. The transition between these two regimes occurs when $\gamma_d \approx 8h_x^2/J$. For the value of $h_x/J = 0.8$ used in our simulations the quantum Zeno regime is for: $\gamma_d \gg 5J$.

## 4.2 Correlation functions

Previously we saw that the dynamics of the magnetization can be described by an intermediate time regime where one can observe the FVD, and a long-time regime describing the thermalization of the system. Next we consider how these two regimes are present in the equal-time two-point connected correlation function, $C(r, t)$:

$$C(r, t) = \frac{1}{N_r} \sum_{i=1}^{L} \left( \langle \sigma_i^z \sigma_{i+r}^z \rangle_t - \langle \sigma_i^z \rangle_t \langle \sigma_{i+r}^z \rangle_t \right). \tag{23}$$

In Eq. (23) we also average over all positions $i$, thus we have introduced a factor of $N_r$ to count all possible pairs of sites separated by a distance $r$. The results of the numerical simulation for the connected correlation function are shown in Fig. (7) for various values of $\gamma_d/J$. When $\gamma_d = 0$, we observe that the correlations grow balistically, after an initial transient that last up to $Jt \simeq 1$. At larger times $Jt \approx 10$ the correlations reach a maximum range, and then begins to turn back. This is related to the confinement of excitations due to the longitudinal magnetic field.

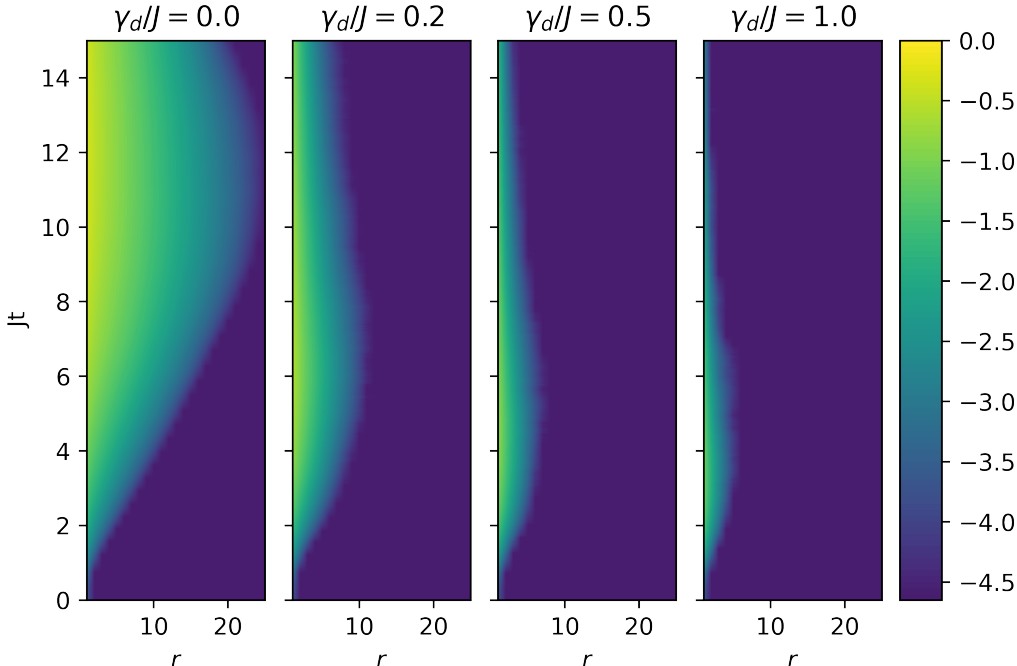

Figure 7: Logarithm of the connected correlation function, Eq. (23), for various $\gamma_d/J$. In the absence of measurements, $\gamma_d = 0$, there is a clear growth of correlations with time due to the unitary dynamics. For finite $\gamma_d$, there is a competition between the fore mentioned unitary dynamics, and dissipation. The measurements decrease the correlations at both large distances and times, in comparison to the closed system.

The presence of continuous measurements progressively kills such correlations, especially in the long-time limit. For small values of $\gamma_d/J$, one can still see that the correlations expand ballistically, but then decay at large values of $r$ and $Jt$. This effect becomes more extreme as one increases the measurement rate, $\gamma_d$, drastically restricting the range (both in space and time) of quantum correlations.

To examine this more carefully, in Fig. (8) we plotted the connected correlation function as a function of $Jt$ at fixed $r = 1$ and as a function of $\gamma_d$. After some initial growth due to the unitary dynamics, there is an exponential decay in the correlations. This exponential decay is evident for all values of $\gamma_d$. The same behaviour can also be shown if one examines the connected correlation function as a function of $r$ for fixed $Jt$, where one observes an exponential decay of the correlations in space, see Fig. (8) b). This decay of correlations is a precursor to the eventual thermalization of the system, and is markedly different from the case $\gamma_d = 0$. Indeed we know that, for any finite measurement rate, $\gamma_d > 0$, the system will asymptotically approach $\rho_{ss}$ which implies

$$\lim_{t\to\infty} C(r, t) = 0, \quad \forall\, r, \tag{24}$$

since the steady-state is completely factorizable in space: $\rho_{ss} = \bigotimes_{i=1}^{L} \mathbb{I}_i/2$, where $\mathbb{I}_i$ is the local $2x2$ identity matrix.

## 4.3 Entanglement entropy

In order to further characterize the behavior of correlations we have also studied the dynamics of the entanglement entropy, Eq. (11). For simplicity we only consider the bipartite entanglement entropy where we trace over half the system.

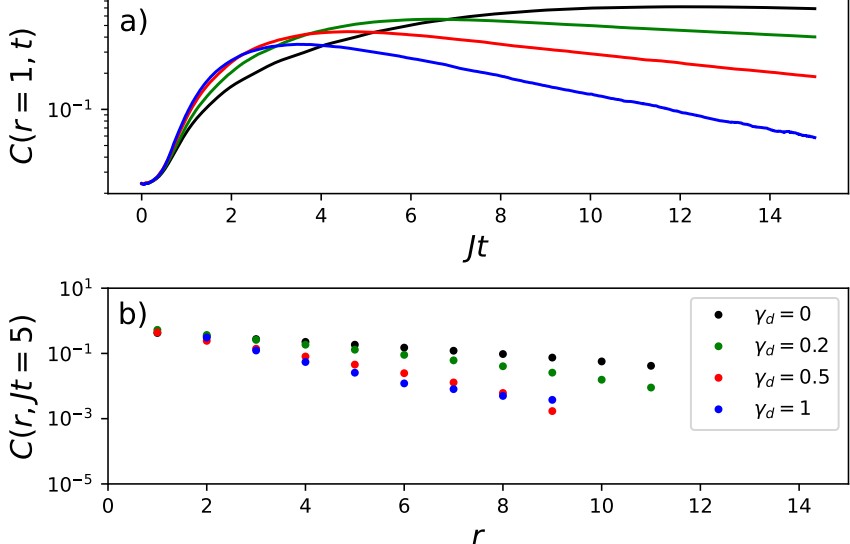

Figure 8: Connected correlation function, Eq. (23), as a) function of $Jt$ at fixed $r = 1$ and b) a function of $r$ at fixed $Jt = 5$ for various $\gamma_d/J$. When $\gamma_d \neq 0$, there is a clear exponential decay. The decay rate of the correlation function as a function of $Jt$ and $r$ depends on $\gamma_d$ and increases with increasing $\gamma_d$.

The entanglement entropy is presented in Fig. 9 for the same parameters as our QT simulations of the magnetization. In the absence of dissipation the entanglement strictly grows and we observe: $S \propto t$ at long times. For small values of $\gamma_d/J$, the entropy still grows linearly in time for $Jt < 15$, however the rate of entropy growth decreases as the measurements destroy the correlations generated by the unitary dynamics. In the time-window observed, this process seems to be non-monotonic with the strength of $\gamma_d$. At short times and large $\gamma_d$, the continuous measurements are more effective at destroying correlations than the unitary dynamics are at increasing them. This tends to lower the entropy initially, and can be readily seen by examining the entanglement entropy via the non-Hermitian Hamiltonian.

It appears that for small values of $\gamma_d$, the time range probed in our simulation belongs to a transient regime, as the entropy does not saturate. The actual duration of this regime is hard to quantify as the unitary and measurement dynamics are competing on equal footing. When $\gamma_d > 0.5J$ we instead see that dynamics due to the measurements overcome the unitary dynamics. For such values of $\gamma_d$ the entropy approaches a stationary state value that decreases monotonically with increasing $\gamma_d$.

We expect such a saturation of the entanglement entropy to occur when the system thermalizes. However, only for $\gamma_d/J \approx 1$ can we observe such physics in the time frame which is accessible to the numerics. As discussed previously, this is because either a) the thermalization time is too long to be observed in our numerics for $\gamma_d/J \ll 1$, or b) we enter the quantum Zeno regime where the approach to the thermal state again becomes too slow to be observed for $\gamma_d/J \gg 1$.

Finally, for $\gamma_d/J = 1$ we show how the entanglement-entropy in the final steady state satisfies an area law. This is evident in Fig. (10), where we find the entanglement entropy to be independent of $L$, up to fluctuations in the trajectories. Such an area law is expected in the quantum Zeno regime. Although we are not strictly in the quantum Zeno regime, we still observe an area law. This is most likely due to the fact the relevant states of the system probed the QTs are those with area law behaviours.

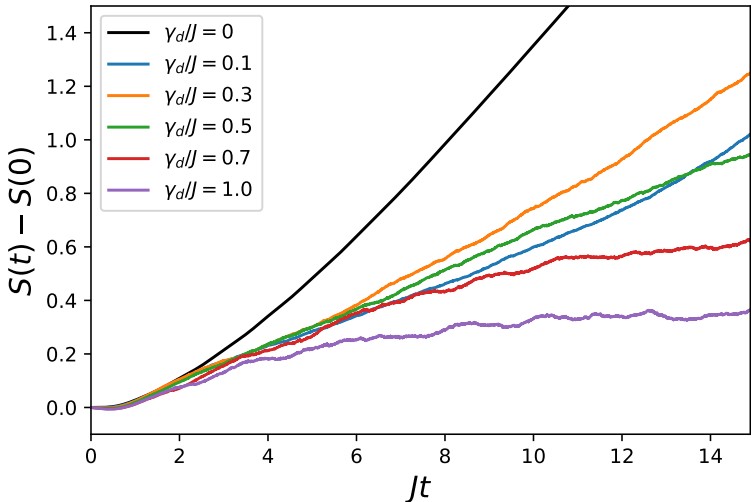

Figure 9: Bipartite entanglement entropy as a function of time for various values of $\gamma_d$. Again we simulate the dynamics using QT with $L = 100$, $h_x/J = 0.8$, and $h_z/J = 0.08$. The statistical error (13) at short times is such that $\sigma(t)/[S(t) - S(0)] \lesssim O(10^{-2})$ while increases at large times $\sigma(t)/[S(t) - S(0)] \lesssim O(10^{-1})$.

## 5 Conclusions

In this work we characterized the decay from the false vacuum of the quantum Ising model in the presence of a measurement apparatus monitoring the local magnetization. To simulate the system dynamics we employed a Monte Carlo matrix-product-state approach. The many-body wavefunction is thus encoded in a matrix-product-state ansatz which evolves in time accordingly to a stochastic Schrödinger equation describing quantum jump trajectories. This protocol allows for the simulation of the real-time dynamics of individual quantum trajectories.

We find that the presence of the continuous monitoring affects the decay of the false vacuum, introducing novel decay paths. In particular, the measurements can locally nucleate spins aligned along the $+z$ and accelerate the departure from the false vacuum. We quantify this process and show that the magnetization fidelity, Eq. (18), decays exponentially within a time window that depends on the measurement rate, and at a rate that is itself exponentially small in the measurement rate.

At long times the system eventually approaches a thermal regime where the mean state of the system is maximally mixed. The typical timescale characterizing the asymptotic approach to the steady state depends on the measurement rate and shows signatures of the quantum Zeno effect. We connect the emergence of this regime to the behaviour of the spectral gap of the Liouvillian and we develop an analytical approach (based on the dissipative Schrieffer-Wolff transformation) able to predict such the Zeno decay rate as well as the critical point.

From the methodological point of view, this work highlights the high potentiality of Monte Carlo matrix product states for the simulation of metastable phenomena in monitored interacting spin systems. This aspect paves the way for more general future explorations concerning, for example, the study the dynamics of the entanglement under different measurement protocols (from quantum jumps to quantum state diffusion [50, 60]) in matrix-product simulations [43].

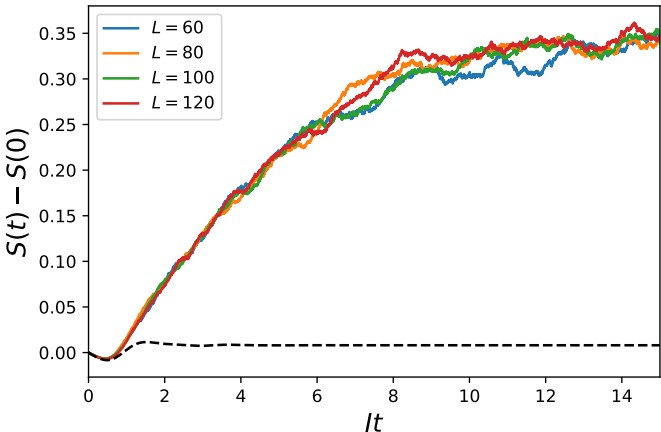

Figure 10: Entanglement entropy, $S(t)$, for fixed $\gamma_d = 1$ and variable system size $L$. When the system thermalizes, the entropy saturates at a value that only depends on $\gamma_d$, not the system size. In this simulation we used $N_{traj} = 600$. We notice that the reduction of the entanglement entropy at short times is due to the purely non-Hermitian evolution (black dashed line) dominating the dynamics at short times.

This work also proposes another avenue for observing the FVD and thermalization in interacting systems. The continuous monitoring can speed up both the FVD and thermalization in a controllable way, rendering them visible on computational and experimental time scales. Although the FVD decay in spin-chains have not been currently observed experimentally, trapped ion experiments can already study non-integrable dynamics of meson confinement [61]. Another potential platform for studying the FVD is the two-component Bose-Einstein condensates [62–65]. There the spin-degrees of freedom act as a quantum Ising model, but with the added benefit of the long coherence time provided by condensates. Extending such studies to optical lattice systems could then lead to direct realizations of similar physics studied in this manuscript.

Finally we note that there are many other intriguing research directions. First and foremost, it would be interesting to develop an analytic treatment of the measurement apparatus via perturbation theory in the regime of small measurement rates, $\gamma_d$. In particular, this could be done for the the dynamics of the mean state by examining the Linbdlad master equation. Beyond this, there are more general questions pertaining the FVD in open quantum systems; such as how much does the FVD physics depend on the different unravelings (corresponding to different measurement protocols) of the Lindblad master equation and on the symmetries of the Hamiltonian.

# Acknowledgements

We acknowledge useful discussions with A. Bastianello, L. Mazza, F. Minganti, D. Rossini, L. Rosso, M. Schiró and S. Scopa. We also acknowledge continuous insightful discussions with the experimental team at the Pitaevskii BEC center (R. Cominotti, G. Ferrari, G. Lamporesi, C. Rogora and A. Zenesini) and theoreticians (A. Recati, G. Rastelli) working on related topics.

**Funding information** We acknowledge financial support from the Provincia Autonoma di Trento.

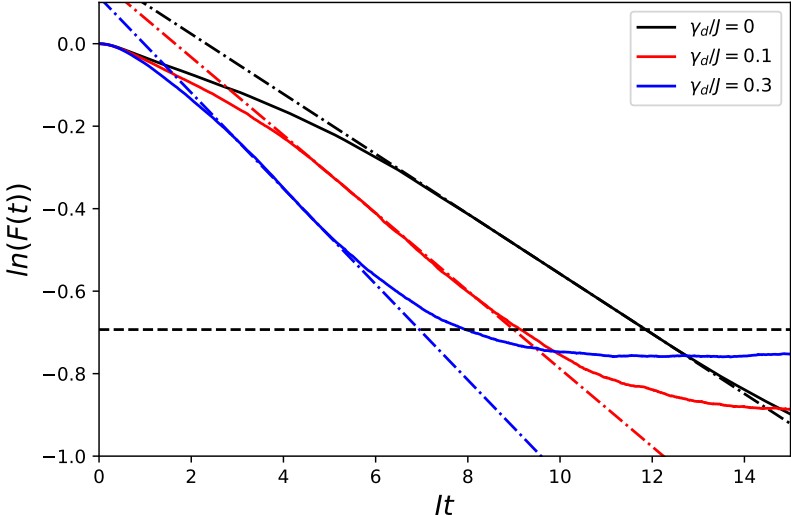

Figure 11: Fitting Protocol for the FVD rate from $F(t)$, Eq. (18). The time-window where the FVD rate is unambiguous is shown in Tab. (1). Within the given domain we fit $\ln(F(t))$ with a linear function, the slope of which is the FVD rate, $\gamma$. The linear fits are shown by the dotted-dashed line, while the solid lines are the results of our numerical simulation.

## A   Details on determining the FVD rate

We determine the FVD rate by examining the dynamics of the magnetization, as shown in Fig. (3). As stated in the main text, the signature of the FVD is an exponential decay away from the initial state, with a decay rate $\gamma$. To emphasize the fitting procedure we plot $\ln(F(t))$ for $\gamma_d/J = 0, 0, 1, 0.3$, where $F(t)$ is the magnetization fidelity defined in Eq. (18). As $\gamma_d$ increases, the time-window where the FVD is observable becomes smaller. In Tab. (1) we show the relevant time-window for various values of $\gamma_d$. These time-windows are only approximate, and we fit the dynamics of $\ln(F(t))$ to a linear fit within these time windows. This procedure appears to be accurate, as extrapolating said fit to the entire FVD regime provides excellent agreement. Examples of this procedure are shown in Fig. (11). The linear fits are shown by the dotted-dashed lines, while the solid lines are the results of the numerical simulation. Within the concerned time-domain, the linear fit, i.e. the exponential decay, is a good description of the dynamics.

Table 1: Time window where the FVD is observed in the dynamics of $F(t)$, Eq. (18). These bounds are only approximate, and the fitting to the FVD is done within these time-domains.

| $\gamma_d$ | $Jt_{min}$ | $Jt_{max}$ |
|---|---|---|
| 0 | 8 | 13 |
| 0.1 | 5 | 7 |
| 0.3 | 4 | 5 |

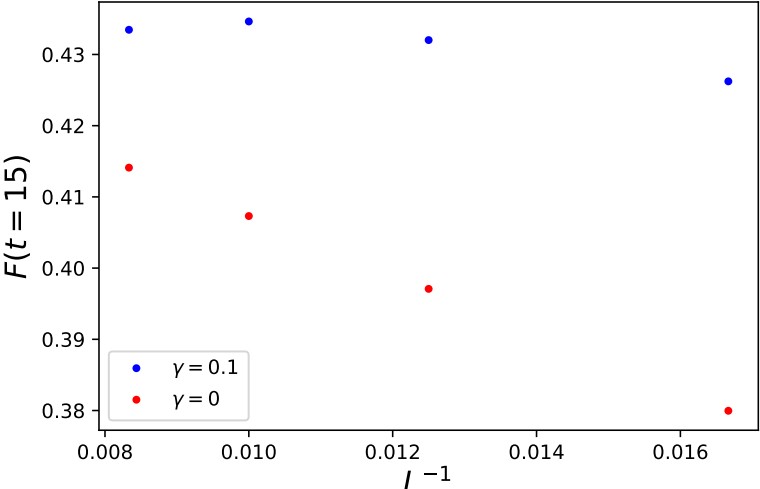

Figure 12: Value of magnetization fidelity, $F(Jt = 15)$, at a given time $Jt = 15$ and for various system sizes $L$ in the presence (blue) and absence (red) of monitoring. In the absence of monitoring, the system is more sensitive to finite size effects. For $\gamma_d = 0.1$ we already see convergence to the infinite size limit for $L = 100$ sites.

## B  Finite size effects

In this work we focus on systems with $L = 100$. It is then natural to ask whether the system is truly in the thermodynamic limit? We examined this issue by looking at both the magnetization, or more exactly $F(t)$ in Eq. (18), at a time $Jt = 15$, both in the presence and absence of dissipation. The results are shown in Fig. (12). From Fig. (12) one can conclude that finite size effects are more important in the absence of continuous monitoring. In our simulations we work at $L = 100$ sites, and one does not see a direct convergence to the thermodynamic limit for $\gamma_d = 0$. For $\gamma_d = 0.1$, we see that the system approaches the thermodynamic limit for $L \approx 100$ sites. This observation is quite natural; in the absence of unitary dynamics correlations can spread throughout the whole system, while the presence of monitoring will kill correlations, especially at larger distances.

This lack of finite size effects in the presence of continuous monitoring can also be demonstrated by considering the entanglement entropy when the system has thermalized. We demonstrated this fact by evaluating the entanglement entropy for various $L$ when $\gamma_d = 1$, as shown in Fig. (10).

## C  The melting of order in the transverse Ising model

In Sec. (4) we considered the FVD dynamics of the quantum Ising model, Eq. (1), in the presence of a finite longitudinal field. As discussed in the main text, the presence of measurements can nucleate single site bubbles of the true vacuum at a rate $\gamma_d$. This mechanism is quite different than that of the closed quantum system, and suggests that one can observe metastability and the melting of order in the transverse Ising model, i.e. when the longitudinal field is zero.

We confirmed this numerically by performing our stochastic matrix product state algorithm on a system of $L = 100$ sites for $h_x = 0.8$ and $h_z = 10^{-4}$. The finite value of $h_z$ was chosen to guarantee convergence to the desired ground state, but is otherwise negligible. The results

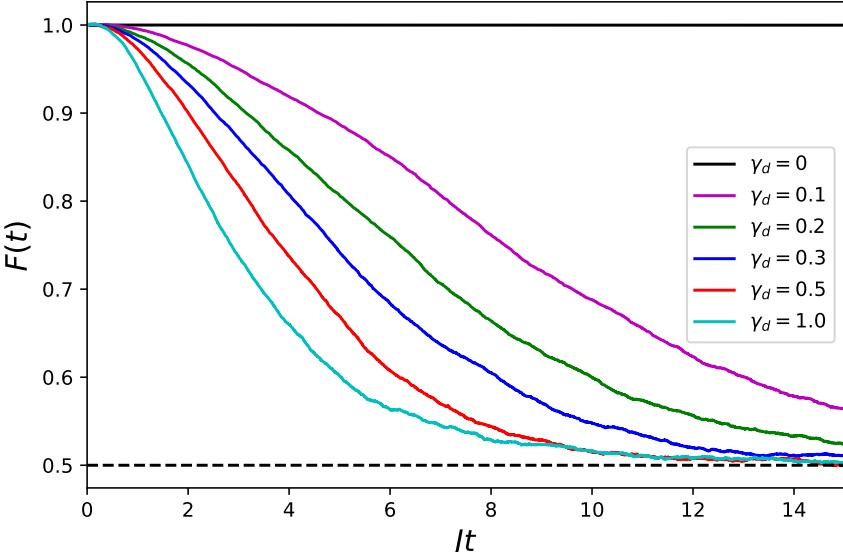

Figure 13: F(t) defined in Eq. (18) for various values of the coupling to the environment, $\gamma_d$, in the absence of the longitudinal field, $h_z = 0$. In these simulations we set $L = 100$ and $h_x = 0.8$, and $N_{traj} = 200$. The dashed line represents the infinite temperature steady state with zero magnetization, i.e. $F(t) = 1/2$.

of our simulations for $N_{traj} = 200$ QTs are shown in Fig. (13). In the absence of measurements, the initial state is in an exact eigentstate of the system, hence there is no evolution of the magnetization. In the presence of measurement, the magnetization decays in a manner qualitatively similar to the case of finite longitudinal field, see Fig. (3).

Similar to the case of finite $h_z$, we can identify a regime where there is an exponential decay away from the initial state with a rate which we also call $\gamma$. Similar to the FVD, $\gamma$ sets the rate at which the initial magnetic order is melted by measurements. We expect $\gamma$ to still obey an Arrhenius law, i.e. Eq. (19) but with $h_z = 0$. To test this we fit the measured decay rates to an Arrhenius law. To simplify the fitting we consider: $\gamma_d \ln(\gamma)$. This transforms the Arrhenius law to a linear fit which is shown in Fig. (14). The linear fit reproduces the data quite well.

# D  Schrieffer-Wolff transformation and its application to the quantum Ising model

In this section we consider the Schrieffer-Wolff transformation for open quantum systems [59], and apply this approach to the open quantum Ising model with both longitudinal and transverse magnetic fields in order to understand the quantum Zeno effect and thermalization time scale.

## D.1  Schrieffer-Wolff transformation for open quantum systems

The Schrieffer-Wolff (SW) transformation is a perturbative approach to generate an effective Hamiltonian or equation of motion for a reduced subspace of relevance to the problem. This can be done to arbitrary order in the coupling of the reduced subspace to the remaining Hilbert space [66]. Here we apply a similar procedure but to the Linblad master equation:

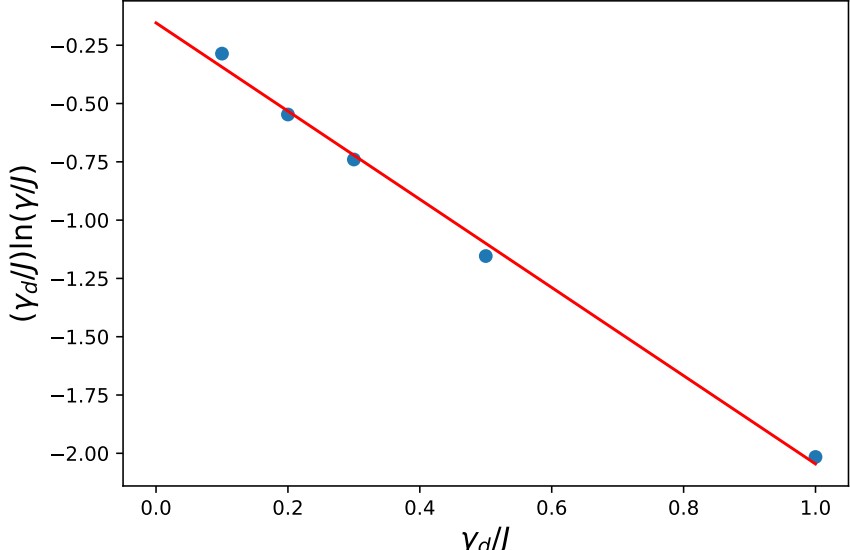

Figure 14: Arrhenius law behaviour for the decay rate $\gamma$, as a function of $\gamma_d/J$. The data corresponds to the results presented in Fig. (13). The red line corresponds to an Arrhenius law, with $h_z = 0$, see Eq. (19).

$$\partial_t \rho(t) = \mathcal{L}\rho(t), \tag{D.1}$$

where $\rho(t)$ is the time-dependent density matrix and $\mathcal{L}$ is the Liouvillian super-operator of the form:

$$\mathcal{L}\rho(t) = -i[H, \rho(t)] + \sum_i \left( L_i \rho(t) L_i^\dagger - \frac{1}{2}\left\{ L_i^\dagger L_i, \rho(t) \right\} \right). \tag{D.2}$$

Eq. (D.2) depends on the many-body Hamiltonian, $H$, and the jump operators $L_i$ which induce dephasing in the system. For the moment we will consider general jump operators and a general Hamiltonian.

As stated previously, Eq. (D.2) is a super-operator, i.e. it maps an operator onto another operator, similar to how an operator maps one state onto another. In this way we can introduce a Hilbert space of all density matrices $\rho(t)$, which the super-operator acts on. A state in this Hilbert space can be represented as a column vector, while the super-operator can be represented as a matrix. This is known as the vectorized representation.

Consider a Liouvillian, $\mathcal{L}_0$. In general $\mathcal{L}_0$ is non-Hermitian and can have complex eigenvalues, These eigenvalues, $\lambda_{\alpha,j}$, can be organized into sectors, $\alpha$, where each eigenvalue in that sector, $j$, is closely spaced. More plainly, the eigenvalue spacing between the sectors, $\lambda_{\alpha+1,j} - \lambda_{\alpha,j}$, is much larger than the spacing within each sector, $\lambda_{\alpha,j+1} - \lambda_{\alpha,j}$. Without loss of generality, we will consider $\alpha = 0$ as the lowest lying eigenvalues of the Liouvillian, while the other sectors have larger eigenvalues. The left and right eigenstates (or rather eigenmatrices) corresponding to these eigenvalues are:

$$\mathcal{L}_0|\alpha, v_j\rangle = \lambda_{\alpha,j}, \qquad \langle\alpha, u_j|\mathcal{L}_0 = \lambda_{\alpha,j}, \tag{D.3}$$

which satisfy the normalizaiton condition:

$$\langle\alpha, u_j|\beta, v_k\rangle = \delta_{\alpha,\beta}\delta_{j,k}. \tag{D.4}$$

Finally we note that the projector onto the $\alpha$ subspace can be written as:

$$P_\alpha = \sum_j |\alpha, v_j\rangle\langle\alpha, u_j|. \tag{D.5}$$

The goal of the SW transformation will be to perturbatively integrate the couplings between each subspace and to construct to construct an effective Liouvillian that is "block diagonal", i.e. no coupling between sectors of differing $\alpha$:

$$\mathcal{L}_{eff} = \sum_\alpha P_\alpha \mathcal{L}_{eff} P_\alpha. \tag{D.6}$$

In this way we can trivially trace out the irrelevant degrees of freedom to a problem.

To this end consider the Liouvillian:

$$\mathcal{L} = \mathcal{L}_0 + \xi\mathcal{L}_1, \tag{D.7}$$

where $\xi$ is a small dimensionless number. In order to implement the SW transformation we use the following transformation:

$$\mathcal{L}' = Q\mathcal{L}Q^{-1}, \tag{D.8}$$

where the operator $Q$ is given by

$$Q = e^\eta, \qquad Q^{-1} = e^{-\eta}. \tag{D.9}$$

Eq. (D.8) is a similarity transformation which preserves the trace of the density matrix. Formally speaking Eq. (D.8) can be written as a set of nested commutators:

$$\mathcal{L}' = \mathcal{L} + [\eta, \mathcal{L}] + \frac{1}{2!}[\eta, [\eta, \mathcal{L}]] + \dots \tag{D.10}$$

Eq. (D.10) can be evaluated to each order in $\xi$ by also expanding $\mathcal{L}'$ and $\eta$ to the appropriate order:

$$\begin{aligned} \mathcal{L}' &= \mathcal{L}^{(0)} + \xi\mathcal{L}^{(1)} + \xi^2\mathcal{L}^{(2)} + \dots, \\ \eta &= \xi\eta^{(1)} + \xi^2\eta^{(2)} + \dots \end{aligned} \tag{D.11}$$

At order $O(\xi^0)$ the effective Liouvillian is just $\mathcal{L}_0$. At $O(\xi)$ one finds:

$$\mathcal{L}^{(1)} = \mathcal{L}_1 + [\eta(1), \mathcal{L}_0], \tag{D.12}$$

As the goal of the SW transformation is to integrate out, i.e. decouple, the subspace $\alpha = 0$ from the higher $\alpha \neq 0$ subspaces, we require that such "off-diagonal" elements vanish. That is we require:

$$\langle\alpha u_k|\mathcal{L}^{(1)}|\beta v_j\rangle = 0, \qquad \alpha \neq \beta. \tag{D.13}$$

From Eqs. (D.12-D.13) one can then obtain the matrix elements for $\eta^{(1)}$:

$$\langle\alpha, u_k|\eta^{(1)}|\beta, v_j\rangle = \frac{\langle\alpha, u_k|\mathcal{L}_1|\beta, v_j\rangle}{\lambda_{\alpha,k} - \lambda_{\beta,j}}, \tag{D.14}$$

for $\alpha \neq \beta$. For $\alpha = \beta$ we can choose: $\langle \alpha, u_j | \eta^{(1)} | \alpha, v_k \rangle = 0$, without loss of generality.

Given Eq. (D.14), one can evaluate the leading correction to the Liouvillian is:

$$\mathcal{L}^{(1)} = \sum_\alpha P_\alpha \mathcal{L}_1 P_\alpha , \tag{D.15}$$

where we have used the property that $\eta^{(1)}$ only couples different sectors of eigenvalues together.

At $O(\xi^2)$ one finds a similar expression for the effective Liouvillian:

$$\mathcal{L}^{(2)} = \left[ \eta^{(1)}, \mathcal{L}_1 \right] + \left[ \eta^{(2)}, \mathcal{L}_0 \right] + \frac{1}{2} \left[ \eta^{(1)}, \left[ \eta^{(1)}, \mathcal{L}_0 \right] \right] . \tag{D.16}$$

Similar to the linear order case, we can again look at the matrix elements of Eq. (D.16). Just as in the first order case, we set the "off-diagonal" matrix elements of Eq. (D.16) to zero, and solve for $\eta^{(2)}$.

To simplify the calculation we note that from Eq. (D.12):

$$\left[ \eta^{(1)}, \left[ \eta^{(1)}, \mathcal{L}_0 \right] \right] = \left[ \eta^{(1)}, \mathcal{L}^{(1)} - \mathcal{L}_1 \right] . \tag{D.17}$$

Thus:

$$\langle \alpha u_k | \eta^{(2)} | \beta v_j \rangle = \frac{1}{2} \frac{1}{(\lambda_{\alpha,k} - \lambda_{\beta,j})} \langle \alpha, u_k | \left( | \left[ \eta^{(1)}, \mathcal{L} \right] + \left[ \eta^{(1)}, \mathcal{L}^{(1)} \right] \right) | \beta, v_j \rangle , \tag{D.18}$$

valid for $\alpha \neq \beta$. A similar analysis to the linear case shows that again $\eta^{(2)}$ only couples different sectors together, hence we only need to consider matrix elements with $\alpha \neq \beta$.

Eq. (D.18) allows one to evaluate the effective action at quadratic order:

$$\mathcal{L}^{(2)} = \frac{1}{2} \sum_\alpha P_\alpha \left[ \eta^{(1)}, \mathcal{L}^{(1)} \right] P_\alpha . \tag{D.19}$$

In terms of the original Liouvillian, the matrix elements of the new effective Liouvillian are:

$$\begin{aligned}
\langle \alpha u_k | \mathcal{L}_{eff} | \alpha v_j \rangle = {} & \langle \alpha u_k | \mathcal{L}_0 | \alpha v_j \rangle + \xi \langle \alpha u_k | \mathcal{L}_1 | \alpha v_j \rangle \\
& + \frac{\xi^2}{2} \sum_\beta \sum_\ell \langle \alpha u_k | \mathcal{L}_1 | \beta v_\ell \rangle \langle \beta u_\ell | \mathcal{L}_1 | \alpha v_j \rangle \left( \frac{1}{\lambda_{\alpha,k} - \lambda_{\beta,\ell}} + \frac{1}{\lambda_{\alpha,j} - \lambda_{\beta,\ell}} \right) .
\end{aligned} \tag{D.20}$$

Eq. (D.20) is the final result which tells one how to construct an effective Liouvillian of the form Eq. (D.6).

## D.2 Application to the quantum Ising model

Let us now consider the application of the SW transformation to the study of the quantum Zeno effect and the thermalization of a one dimensional quantum Ising model with transverse and longitudinal magnetic fields that is coupled to an infinite thermal bath.

The dynamics of the density matrix are governed by Eq. (D.2). The unitary dynamics are governed by the following Hamiltonian:

$$H = -\sum_i \left[ J \hat{\sigma}_i^z \hat{\sigma}_i^z + h_x \hat{\sigma}_i^x + h_z \hat{\sigma}_i^z \right] , \tag{D.21}$$

where $\hat{\sigma}_i^{(x,y,z)}$ is the $x,y,z$ Pauli matrix for the $i=1,2,...N$ site. The dephasing is governed by the set of jump operators for each site $i$:

$$L_i = \sqrt{\gamma_d}\frac{1}{2}\left(\hat{I}_i + \hat{\sigma}_i^z\right),\tag{D.22}$$

with $\gamma_d$ as the dephasing rate and $\hat{I}_i$ is the identity operator for site $i$.

In the quantum Zeno limit, $J \ll \gamma_d$, the unitary part of the Liouvillian acts as a small perturbation. Hence we define the zeroth-order Liouvillian as:

$$\mathcal{L}_0\rho(t) = \sum_i \left(L_i\rho(t)L_i^\dagger - \frac{1}{2}\{L_i^\dagger L_i, \rho(t)\}\right).\tag{D.23}$$

It is straightforward to show that Eq. (D.23) has a set of states with zero-eigenvalue, the so-called dark states. These states do not exhibit dissipation and have $\lambda_{0,j} = 0$. These states are associated with the probability density matrices:

$$|0,j\rangle = |\{\sigma_i\}\rangle\langle\{\sigma_i\}|,\tag{D.24}$$

where $|\{\sigma_i^z\}\rangle$ is a many-body state with definite spin along the z-direction:

$$\sum_i \hat{\sigma}_i^z|\{\sigma_i^z\}\rangle = \sum_i \sigma_i|\{\sigma_i^z\}\rangle,\tag{D.25}$$

with $\sigma_i = \pm 1$.

The next degenerate set of states have eigenvalues $\lambda_{1,j} = -\gamma_d/2$ and corresponds to density matrices of the form:

$$|1,i\rangle = |\{\sigma_i\}\rangle\langle\{\sigma_i\}'|,\tag{D.26}$$

where $|\{\sigma_i\}'\rangle$ denotes a many-body state that differs from $|\{\sigma_i\}\rangle$ by a single flipped spin.

The unitary evolution will naturally couple these sets of eigenstates together. Thus we treat:

$$\mathcal{L}_1 = -i[H_x, \rho(t)],\tag{D.27}$$

where $H_x = -\sum_i h_x\hat{\sigma}_i^x$ is the contribution to the Hamiltonian from the transverse field. Then we can apply the derived SW transformation to obtain an effective theory for the dark states. Before proceeding further we note that in writing Eq. (D.27) we note that the remaining terms of the Hamiltonian in Eq. (D.21) produce a vanishing result.

Upon substituting Eq. (D.27) into Eq. (D.20), on can immediately show that the term linear in $h_x$ is zero and one needs to go to quadratic order. A careful examination of the matrix elements shows that one can write the effective Liouvillian at second order in $h_x$ as:

$$\mathcal{L}_{eff} = \frac{4h_x^2}{\gamma_d}\sum_i\left[\hat{\sigma}_i^x\rho_0(t)\hat{\sigma}_i^x - \rho_0(t)\right],\tag{D.28}$$

where $\rho_0(t) = P_0\rho(t)P_0$ is the density matrix projected onto the set of dark states. Eq. (D.28) has the form of a Liouvillian with no unitary time evolution, but with dissipation in the $x$-direction with strength, $4h_x^2/(\gamma_d)$.

It is well known that the system will approach its steady state on a time scale set by the Liouvillian gap which in our case is simply the negative of the smallest finite eigenvalue of the effective Liouvillian super-operator. This is because the Liouvillian gap represents the longest time scale in the problem, while the larger eigenvalues of the Liouvillian represent motion that have been damped out. Given Eq. (D.28) it is straightforward to show 1) the the steady state is still the maximally mixed state, i.e. a state with infinite temperature, and 2) that the Liouvillian gap is $8h_x^2/\gamma_d$, or equivalently the thermalization time scale, $\tau_{th}$, is:

$$\tau_{th} = \frac{\gamma_d}{8h_x^2} \, . \tag{D.29}$$

The linear dependence of $\tau_{therm.}$ on $\gamma_d$ in this regime is indicative of the quantum Zeno effect. Increasing the dissipation slows down the dynamics as the thermalization is ultimately controlled by states that are dark to the dissipation.

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
