# Peer review of "Monte Carlo matrix-product-state approach to the false vacuum decay in the monitored quantum Ising chain"

_SciPost Physics, doi:SciPost Phys. 15, 152 (2023)_

## Round 1 · Referee Report · Anonymous (Referee 1) · 2023-7-8

Report

The authors present a numerical study of the false vacuum decay in the ferromagnetic quantum Ising chain subject to continuous monitoring of the local magnetization. They investigate the competition between coherent dynamics, which leads to the creation of resonant bubbles of the true vacuum, and measurements, which induce heating and reduce quantum correlations. The authors use a numerical approach based on the combination of matrix product states with stochastic quantum trajectories to simulate the trajectory-resolved non-equilibrium dynamics of the system. They analyze the effects of continuous monitoring on the false vacuum decay and thermalization using quantities such as magnetization, correlation functions, and trajectory-resolved entanglement entropy. Overall, the paper is well-written and presents an interesting investigation of the false vacuum decay in the monitored quantum Ising chain. The authors provide a thorough introduction to the topic, describe the model and measurement scheme, explain their simulation protocol, present their results, and draw conclusions based on their findings. The paper is organized and structured in a clear and logical manner.

Below are my specific comments:

  • The authors could provide further clarification or discussion on why they chose the specific jump operators used in their simulations. My understanding is that the protocol with the projection to the up state is designed to enhance bubble nucleation. Would you expect a similar effect if one chooses, for example, to project in the down state or in the + or - states? Is it sufficient to heat up the system to enhance the decay or do you expect the specific type of measurement to play a role?

  • While I understand the discussion on quantum trajectories and the procedure used for their simulation, I found the explanation of Eq. 4 very confusing. It would be useful to explain more clearly what the vector $N_t$ is and how the state $\psi(N_t)$ depends on it.

  • Can the non-monotonicity of the rate of growth of entanglement entropy be attributed to the non-Hermitian term that slows down the evolution for small $\gamma_d$?

  • Are the statistical error bars negligible when averaging over different trajectories (for example in Fig. 3, 8, 9, 10)?

  • I suggest making some changes to improve the visualization of the data. For Figure 8b, the general trend is hard to observe. It would be useful to change the aspect ratio by reducing the width and to adjust the position of the legend so that it does not cover the data. For Figure 9, it is not easy to distinguish saturation from slow growth. I suggest reducing slightly the range of the y-axis and maybe adding an inset with a zoom at low values of $S(t)-S(0)$.

Requested changes

  1. Discussion on the choice of measurement protocol.
  2. Clarification on Eq. 4.
  3. Changes on Figures for easier visualization.

  • validity: high
  • significance: good
  • originality: good
  • clarity: good
  • formatting: good
  • grammar: good

Author:  Jeffrey Allan Maki  on 2023-08-18  [id 3911]

(in reply to Report 1 on 2023-07-08)
Category:
answer to question

We thank the referee for his/her positive and critical view of our manuscript. Here we reply point-by-point to issues raised in the review.

The referee writes:

The authors could provide further clarification or discussion on why they chose the specific jump operators used in their simulations. My understanding is that the protocol with the projection to the up state is designed to enhance bubble nucleation. Would you expect a similar effect if one chooses, for example, to project in the down state or in the + or - states? Is it sufficient to heat up the system to enhance the decay or do you expect the specific type of measurement to play a role?

Our response:

In this work we are interested in studying how the local measurement of the order parameter of the model affects the decay of the false vacuum. Given that for our choice of the parameters the false vacuum is the state with all the spin aligned approximately in the −z direction, we chose to measure the +z component of the spin. In this way we can simulate the situation where the local detectors act as a nucleator of local defects of the true vacuum and then addressing their dynamics. If, on the contrary, we chose to measure −z, the dynamics will be drastically different. Since the initial state has a large magnetization along the −z, the act of measurements will be to keep the spins locked in the −z direction. For large measurement rate the system will enter a quantum Zeno regime where the dynamics are basically frozen. We have added a discussion about the choice of jump operators on page 6.

The referee writes:

While I understand the discussion on quantum trajectories and the procedure used for their simulation, I found the explanation of Eq. 4 very confusing. It would be useful to explain more clearly what the vector \({\bf N_t}\) is and how the state \(\psi({\bf N_t})\) depends on it

Our response:

To summarize, the vector \({\bf N_t}\) dictates when and where a quantum jump occurs. Given the vector \({\bf N_t}\) one knows exactly how the quantum state evolved. Specifying \({\bf N_t}\), is equivalent to the knowledge of the wavefunction during a single experimental realization of the system. In the new version of the manuscript we have improved the explanation of \({\bf N_t}\) and the role it has in quantifying the state of the system.

The referee writes:

Can the non-monotonicity of the rate of growth of entanglement entropy be attributed to the non-Hermitian term that slows down the evolution for small \(\gamma_d\)?

Our response:

We thank the referee for this question. The answer is yes. We have investigated this issue and found the dip in the entanglement entropy is in fact related to the non-Hermitian evolution. Indeed the non-Hermitian part of the dynamics (no-click evolution) in general slows down the build up of quantum correlations and competes with the \(h_x\). When \(\gamma_d\) is large the no-click evolution shows a reduction of correlations at short times. We have updated the manuscript accordingly.

The referee writes:

Are the statistical error bars negligible when averaging over different trajectories (for example in Fig. 3, 8, 9, 10)?

Our Response:

Yes there are statistical errors due to the finite number of trajectories employed in the simulations. In the case of the magnetization and correlations they are negligible (of about 0.01% of the signal) while for the entropy they are more relevant (about 5 % of the signal at large times). We have added discussions on how to calculate the statistical variance on page 8, and reported the values where appropriate.

The referee writes:

I suggest making some changes to improve the visualization of the data. For Figure 8b, the general trend is hard to observe. It would be useful to change the aspect ratio by reducing the width and to adjust the position of the legend so that it does not cover the data. For Figure 9, it is not easy to distinguish saturation from slow growth. I suggest reducing slightly the range of the y-axis and maybe adding an inset with a zoom at low values of \(S(t)-S(0)\).

Our Response:

We thank you for your suggestions, and have made several changes to imporve the visualization of the data.

---

## Round 1 · Referee Report · Giulia Piccitto (Referee 2) · 2023-7-11

Strengths

  1. The paper is well organized and well written, especially in the introductory part;
  2. The main topic is in a very active research area;
  3. It proposes a different perspective in the field of monitored systems.

Weaknesses

  1. Some aspects must be discussed more in details, e.g. the connection between the observables used to characterize the false vacuum decay.

Report

In this paper the authors investigate the role of continuous monitoring on the decay of the false vacuum of a quantum Ising chain in presence of both a longitudinal and a transversal field. The simulation of the stochastic dynamics is done by adopting a MPS representation of the many-body state. The main result is that the presence of an external measurement apparatus accelerates the decay to the thermal state. This claim is supported by the analysis of many quantities such as the magnetization fidelity, the local magnetization, the Liouvillian spectrum, the correlation functions and the entanglement entropy.

I find the introductory part of the manuscript well written, while I find the part of the results a bit disorganized. In what follows the main comments.

  1. Can the authors explain why Eq. (8) is equivalent to an infinite temperature bath? Is this obvious for any Hamiltonian and any measurements strength? I agree that the presence of dephasing would lead to decoherence and, consequently, to a diagonal ensemble, but I do not get why this should be the maximally mixed one. My concern is especially for the two cases of a) arbitrary $\gamma_d$ and $h_z=0$, where, because of the integrability of the model, I can expect the elements of the diagonal to have different magnitude (especially for finite sizes); b) the case of the quantum Zeno regime, where the system is confined in a small subregion of the Hilbert space and cannot be described by Eq. (9).

  2. Can the authors comment on the reliability of the Monte Carlo matrix product state? Why is this algorithm suitable for the model under consideration?

  3. In Fig. (3), the curves for small $\gamma_d/J$ goes below the infinite temperature value, suggesting that this is just a transient before the steady state is reached. Have the authors checked that the steady state is actually reached at the expected thermalization time? If this check is numerically achievable, I suggest to add an inset with the late times dynamics. Moreover, looking at Fig. (3) it seems that for large $\gamma_d/J$ the fidelity never goes below the steady state one. Do you have any insight of this?

  4. The major criticism I have regards Section 4 that seems a bit disorganic. As a reader, it is difficult to get the connections between the different observables discussed, i.e. it seems a collection of results. For example, I missed the relation, if any, between $\gamma$ in Eq. (19) and $\gamma_{th}$ in Eq. (21). In other words, are Fig.(4) and Fig.(5) connected? In general, I miss the connection between the values $\gamma_d$ at which any of the quantities of interest exhibit some qualitative change. I think that the paper can improve a lot if a comprehensive discussion of these connections is added (for example at the beginning of the section).

  5. Can the authors comment on the possibilities of considering periodic boundary conditions?

Minors: - Maybe it could be useful to better format the equation $\tilde{\ell} = 1 + m/h_z/M$ , under Eq.(2) - Across Section 4, the quantity introduced in Eq. (17) is referred as "figure of merit" while in the conclusion is referred as "magnetization fidelity". Can I suggest to use the latter in the whole manuscript? - Typo: At the beginning of Sec. 4.3 a bracket is missing "Fig. 7". - Under Eq. (23) you define the stationary state as a tensor product of 2x2 identity matrices labelled as $\mathbb{I}$. This symbol in Eq. (9) was used to identify the $2^N \times 2^N$ identity matrix. Maybe you can choose two different symbols.

Requested changes

  1. Reorganization of Section 4 as discussed in the report;
  2. Adding some comments on the point discussed in the report;
  3. Correct the typos listed in Minors.

  • validity: good
  • significance: good
  • originality: good
  • clarity: good
  • formatting: good
  • grammar: good

Author:  Jeffrey Allan Maki  on 2023-08-18  [id 3912]

(in reply to Report 2 by Giulia Piccitto on 2023-07-11)
Category:
answer to question

We would also like to thank referee 2 for his/her comments and warm reception of our paper. We have made a number of changes in response to your comments and criticisms.

The referee writes:

Can the authors explain why Eq. (8) is equivalent to an infinite temperature bath? Is this obvious for any Hamiltonian and any measurements strength? I agree that the presence of dephasing would lead to decoherence and, consequently, to a diagonal ensemble, but I do not get why this should be the maximally mixed one. My concern is especially for the two cases of a) arbitrary \(\gamma_d\) and \(h_z = 0\), where, because of the integrability of the model, I can expect the elements of the diagonal to have different magnitude (especially for finite sizes); b) the case of the quantum Zeno regime, where the system is confined in a small subregion of the Hilbert space and cannot be described by Eq. (9).

Our response:

We thank the referee for raising this point. We have analysed this issue pertaining to our problem. For finite \(\gamma_d\), the final steady state is always the maximally mixed state. This is because in our model, the presence of measurements along the +z direction induces an effective imaginary longitudinal field (with strength \(-\gamma_d/4\)), even when \(h_z = 0\). This spoils all symmetries of the transverse Ising model leading asymptotically to the diagonal ensemble in the full Hilbert space. We explicitly verified this statement with numerically exact integration of the master equation for small system sizes (not shown) and computing the fidelity of the steady state with the maximally mixed one.

We note that in the quantum Zeno regime, the system still thermalizes to the maximally mixed state, but the thermalization rate is very slow as noticed by the referee. This is illustrated in the new discussions in the main text and in Appendix D.2.

The referee writes:

Can the authors comment on the reliability of the Monte Carlo matrix product state? Why is this algorithm suitable for the model under consideration?

Our response:

We thank the referee for this question. In the framework of monitored many-body systems the aim is to simulate the trajectory-resolved dynamics of extended quantum system in the presence of measurements. While for free Hamiltonians and Gaussian measurement protocols the dynamics can be simulated efficiently within the framework of Gaussian states, the situation dramatically changes when interacting models and/or non-Gaussian measurements are considered. The false vacuum decay in quantum spin chains with continuous measurements of the order parameter falls in this class of problems.

The algorithm we use to tackle this situation is specifically suitable for the model and measurement protocol under consideration. Matrix product states are particularly efficient for the simulation of slightly entangled quantum states in \(D=1\). This is the case for the hybrid dynamics (unitary evolution together with measurements) of our model. On the one hand, the unitary dynamics is ruled by an Hamiltonian having a small integrability breaking term (the longitudinal field proportional to \(h_z\)) which is main responsible for the build up of quantum correlations between different sites. On the other hand, local continuous measurements are easily naturally implemented within this ansatz since the action of local operators (or, alternatively, the evolution with \(H_{\rm eff}\)) takes place through the action of local operators (or complex TEBD evolution). Furthermore, quantum measurements reduces the amount of correlations generated by the unitary evolution and, as result, allow us to simulate trajectory-resolved dynamics up to relatively large times for large system sizes.

The referee writes:

In Fig. (3), the curves for small \(\gamma_d/J\) goes below the infinite temperature value, suggesting that this is just a transient before the steady state is reached. Have the authors checked that the steady state is actually reached at the expected thermalization time? If this check is numerically achievable, I suggest to add an inset with the late times dynamics. Moreover, looking at Fig. (3) it seems that for large \(\gamma_d/J\) the fidelity never goes below the steady state one. Do you have any insight of this?

Our response:

For the large system size considered (\(L=100\)) we are not able to witness the complete thermalization of the system to the maximally mixed state for small values of \(\gamma_d/J\) since the associated timescale is proportional to \(J/\gamma_d\). During the dynamics the trajectories increase their entanglement, as shown in Fig.(9), and at a point we are no longer able to describe the quantum state efficiently within our MPS ansatz. However, as also discussed in the first point of this reply, we know that (a) analytically, the maximally mixed state is the steady state of the problem for and (b) we checked with the exact numerical integration of the master equation (for the values of parameters considered in Fig.(3) that the steady state is indeed the maximally mixed state.

The referee writes:

The major criticism I have regards Section 4 that seems a bit disorganic. As a reader, it is difficult to get the connections between the different observables discussed, i.e. it seems a collection of results. For example, I missed the relation, if any, between \(\gamma\) in Eq. (19) and \(\gamma_{th}\) in Eq. (21). In other words, are Fig.(4) and Fig.(5) connected? In general, I miss the connection between the values \(\gamma_d\) at which any of the quantities of interest exhibit some qualitative change. I think that the paper can improve a lot if a comprehensive discussion of these connections is added (for example at the beginning of the section).

Our response:

We thank the referee for this comment. We have taken this criticism to hart and have reorganized the results section providing a more organic view of our findings. We now primarily focus on the two regimes, an intermediate regime where we observe the false vacuum decay, and a long time regime where we observe the thermalization dynamics. The typical rates in these two regimes, \(\gamma\) and \(\gamma_{\rm th}\) respectively, are not related to one another, other than that they both depend on the continuous measurement. We have now clarified this point in the revised version of the manuscript. Our results sections are restructured along this line.

Minor Changes:

We have also taken care of the minor chnages suggested by the referee.

---

## Round 2 · Referee Report · Anonymous (Referee 3) · 2023-8-18

Report

The authors have addressed my concerns in a comprehensive and satisfactory manner. Their revisions and explanations have improved the clarity and quality of the manuscript. I now recommend publication in SciPost Physics.

---

## Round 2 · Author Response

Dear editors and referees,

We would like to thank you for your careful and critical reading of our manuscript. The feedback has improved the quality of the presentation greatly. We have taken the time to address each and all of the referee concerns. We hope this improved version meets the referees' standards, and we are now resubmitting the manuscript to SciPost Physics.

---

## Round 2 · List of Changes

The following is the list of changes:

-Added a discussion about the choice of jump operators
-Improved the explanation of the uncorrelated Poisson processes and how it parametrize the state of a system for a single experimental realization
- Improved discussion on the steady state
-Added a discussion about the statistical variance and how it depends on the number of trajectories. The calculated variance has been added to the appropriate figures.
-Restructured the results section to improve readability and highlight the connections between results
-Added a discussion about the non-monotonic nature of the entanglement entropy
-Updated figures 8-10
- Improved the discussions on the derivation of the thermalization time scale
-Several minor grammatical and clerical changes
-Added references 26, 27, and 44 of the new manuscript

---

## Editorial Decision

published